# Potential Use of *Quercus dalechampii* Ten. and *Q. frainetto* Ten. Barks Extracts as Antimicrobial, Enzyme Inhibitory, Antioxidant and Cytotoxic Agents

**DOI:** 10.3390/pharmaceutics15020343

**Published:** 2023-01-19

**Authors:** Corneliu Tanase, Mihai Babotă, Adrian Nișca, Alexandru Nicolescu, Ruxandra Ștefănescu, Andrei Mocan, Lenard Farczadi, Anca Delia Mare, Cristina Nicoleta Ciurea, Adrian Man

**Affiliations:** 1Department of Pharmaceutical Botany, Faculty of Pharmacy, “George Emil Palade” University of Medicine, Pharmacy, Sciences and Technology of Târgu Mures, 38 Gheorghe Marinescu Street, 540139 Târgu Mures, Romania; 2Research Center of Medicinal and Aromatic Plants, “George Emil Palade” University of Medicine, Pharmacy, Sciences and Technology of Târgu Mures, 38 Gheorghe Marinescu Street, 540139 Târgu Mures, Romania; 3Department of Pharmaceutical Botany, Faculty of Pharmacy, “Iuliu Hatieganu” University of Medicine and Pharmacy, Gheorghe Marinescu Street 23, 400337 Cluj-Napoca, Romania; 4Doctoral School of Medicine and Pharmacy, “George Emil Palade” University of Medicine, Pharmacy, Sciences and Technology of Târgu Mures, 38 Gheorghe Marinescu Street, 540139 Târgu Mures, Romania; 5Laboratory of Chromatography, Institute of Advanced Horticulture Research of Transylvania, University of Agricultural Sciences and Veterinary Medicine, 400372 Cluj-Napoca, Romania; 6Department of Pharmacognosy and Phytotherapy, Faculty of Pharmacy, “George Emil Palade” University of Medicine, Pharmacy, Sciences and Technology of Târgu Mures, 38 Gheorghe Marinescu Street, 540139 Târgu Mures, Romania; 7Chromatography and Mass Spectrometry Laboratory, Center for Advanced Medical and Pharmaceutical Research, “George Emil Palade” University of Medicine, Pharmacy, Sciences and Technology of Târgu Mures, 38 Gheorghe Marinescu Street, 540139 Târgu Mures, Romania; 8Department of Microbiology, Faculty of Medicine, “George Emil Palade” University of Medicine, Pharmacy, Sciences and Technology of Târgu Mures, 38 Gheorghe Marinescu Street, 540139 Târgu Mures, Romania

**Keywords:** *Q. dalechampii*, *Q. frainetto*, antimicrobial, phenolic compounds, rhytidome

## Abstract

The bark resulted as a by-product after different *Quercus sp.* processing is a valuable alternative source of phenolic compounds (mainly tannins). Hence, the focus of the present work was to obtain eight extracts from the rhytidome of the less-studied *Q. dalechampii* and *Q. frainetto* and characterize them in terms of chemical and bioactive profiles. Ultrasound and microwave-assisted extraction methods were used for the preparation of the extracts. Total phenolic and tannin contents were assessed through classic spectrophotometric methods, while several individual phenolic compounds were identified and quantified using UPLC-PDA. Antioxidant, enzyme-inhibitory, antibacterial, and cytotoxic activities were tested using in vitro assays; additionally being evaluated was the ability of the extracts to inhibit the adherence of MRSA to suture wires. The UPLC analysis confirmed the presence of gallic acid, catechin, taxifolin, vanillic acid, epicatechin, and caffeic acid. The results showed that tested extracts were able to exert cytotoxic effects, at 6% and 3% concentrations, on confluent cells. The tested solutions inhibit α-glucosidase activity and the antibacterial potential suggested a mild to moderate effect against the Gram-positive strains. Overall, the obtained results revealed rich phenolic and tannin contents for the extracts obtained from both species through microwave-assisted extraction, probably responsible for their mild antibacterial and cytotoxic effects.

## 1. Introduction

Phenolic compounds are the most valuable phytochemicals widespread in plant matrices, intensively promoted in the last decades both as part of an equilibrate diet and molecules with preventive and therapeutic potential in the management of various diseases [1,2]. Due to this, actual trends are described by increased demand for phenolic-rich products, especially for the food and pharmaceutical industry, which consequently incited the research for alternative sources for these biocompounds. A suitable solution for this challenge seems to be the use of plant-derived by-products to obtain extracts enriched in phenolic compounds, based on the circular economy and bioeconomy principles, intensively applied in the last years for the management of forest processing waste [3,4,5].

*Quercus* species, commonly known as “oaks”, belong to the group of the most exploited deciduous trees, which are a source of wood for the manufacture of furniture as well as for fuel. The main waste resulting from the primary processing of oak is represented by its bark, which is mostly used as an alternative fuel or tanning agent in the leather industry [6,7]. Moreover, the bark of different oak species is cited as a herbal product with long-term documented use in human and veterinary medicine, especially for their tannin-rich content correlated with astringent, hemostatic, antibacterial, or antidiarrheal properties [8,9,10,11]. Officially, European Pharmacopoeia 10.8 recognizes the herbal product *Quercus cortex,* which consists of the bark collected from the fresh young branches of *Quercus robur* L., *Q. petraea* (Matt.) Liebl. and *Q. pubescens* Willd. with a minimum content of 3% tannins (expressed as pyrogallol) in the dried products; similar regulations are mentioned for the same product in EMA HMPC (Committee on Herbal Medicinal Products) monograph (tannins content between 2–20%) [8,12]. Besides the already proven bioactivities, oak bark extracts were intensively tested in last years for other potential health benefits; a positive impact against oxidative stress biomarkers and the improvement of fatigue syndrome were reported both in pre-clinical and clinical evaluation for Robuvit^®^, a dietary supplement containing an *Q. robur* standardized extract (min. 40% phenolic compouns) [13,14,15], while for the bark extracts of the same species was observed a dose-dependent cytotoxic effect tested in a larvae experimental model [16].

Additionally to the above-mentioned statements, the barks of other *Quercus* species are traditionally used as herbal medicines depending on their occurrence in different geographic areas or through accidental substitution (due to unclear morphological differences between species) [8]. Based on the empirical observations, many researches were conducted in order to unravel the chemical and bioactive profile of other oak species barks [7,9,17]. As well for the well-known *Q. robur*, the bark extracts of *Q. cerris* var. *cerris*, *Q. macranthera* subsp. *syspirensis* and *Q. aucheri* were confirmed as cytotoxic agents against the Hep-2 cell line [18], while a potential antidiabetic potential was proven for *Q. rubra* (an invasive species for European areas) bark extracts through the inhibition of *α*-glucosidase activity [19].

*Q. dalechampii* (Dalechamp oak) an *Q. frainetto* (Hungarian oak) are native to southeastern Europe (including the Balkan peninsula and Turkey), being commonly found in deciduous forests of these areas [20,21,22]. Systematic literature research reveals the lack of consistent data on the chemical and bioactive profile of these species, hence, the aims of the present study were: (1) the characterization of the phytochemical profile of *Q. dalechampii* and *Q. frainetto* rhytidome extracts; (2) the evaluation of their biological activities (antioxidant, antibacterial, antifungal, cytotoxic, and enzyme inhibitory activities); and (3) the assessment of the influence of extraction procedure and extraction parameters against the chemical and bioactive profile of the above-mentioned extracts.

## 2. Materials and Methods

### 2.1. Chemicals, Reagents, and Bacterial Strains

The ethanol used as solvent (95%) was from Girelli Alcool Srl (Zibido, San Giacomo, Italy). Sodium carbonate decahydrate was purchased from Reactivul Srl (Râmnicu, Vâlcea, Romania), gallic acid monohydrate purchased from Sigma-Aldrich Chemie GmbH (Steinheim, Germany), and Folin–Ciocâlteu reagent purchased from Merck KGaA (Darmstadt, Germany). For the assessment of total tannin content and antioxidant activity the following reagents were used: 2,2-diphenyl-1-picrylhydrazyl (DPPH), 2,2′-azino-bis(3-ethylbenzothiazoline-6-sulfonic acid) (ABTS), hide powder, and pyrogallol, all acquired from Sigma-Aldrich Chemie GmbH (Steinheim, Germany).

Additionally, phosphate-buffered saline (PBS), acetylcholinesterase from electric eel (C3389), *α*-glucosidase from *Saccharomyces cerevisiae* (G5003), and tyrosinase from mushroom (T3824), acarbose, galantamine, kojic acid, dimethyl sulfoxide (DMSO), 5,5′-dithiobis(2-nitrobenzoic acid) (DTNB), levodopa (L-DOPA), 4-nitrophenyl-*β*-D-glucopyranoside (*p*-NPG) and Tris base were used for enzyme assays (all from Sigma Aldrich). All other reagents were purchased from local suppliers.

The microorganisms used for antimicrobial evaluations (five bacterial and 3 fungal strains) were supplied through the Microbiology Department of the George Emil Palade University of Medicine, Pharmacy, Sciences, and Technology from Târgu-Mureș.

The standards used for both spectrophotometric and UPLC-PDA analysis (chlorogenic acid, caffeic acid, gallic acid, epicatechin, catechin, eleuteroside B, quercetin, quercetin arabinoglycoside, vanillic acid, taxifolin, sinapic acid, trolox, acarbose, kojic acid, galantamine hydrobromide) were purchased from Sigma-Aldrich.

### 2.2. Plant Material Collection

*Q. dalechampii* and *Q. frainetto* rhytidomes (whole mature bark containing periderm with outter dead tissues) were collected from Deva, Hunedoara County, Romania, in May 2021 by manually shredding from trunks and old branches of 90–100 years old trees; collection was preceded by authentication of species confirmed based on morphological features evaluation (Figure 1), performed by Dr. Corneliu Tanase. In *Q. frainetto* the bark is light gray and cracks into small square plates (Figure 1b,c). *Q. dalechampii* has brown bark with deep fissures (Figure 1a). Further, bark fragments were supposed to drying procedure (artificial drying for 24 h at 50 °C using a Nahita 631 Plus drying oven—Auxilab S.L., Beriáin, Spain) followed by milling of dried plant material (Pulverisette 15 cutting mill—Fritsch GmbH, Idar-Oberstein, Germany) [23,24].

### 2.3. Extraction Procedure

Two extraction methods were applied in order to obtain phenolic-rich extracts from *Q. dalechampii* and *Q. frainetto* barks, namely miccrowave-assisted extraction and ultrasound-assisted extraction. Bothe methods were applied using as solvents water and 70% *v/v* ethanol (70% EtOH).

*Microwave-assisted extraction:* an optimized method previously described was used; 10 g of powdered bark were extracted with 200 mL of solvent using an Ethos X Advanced microwave extractor (Milestone, Sorisole, Italy); for hydroethanolic extraction, extraction occurred for 18 min at an irradiation power of 650W, while for aqueous extraction the equipment was operated at 850 W for 30 min.

*Ultrasound-assisted extraction:* 2.5 g of powdered bark were mixed with 100 mL solvent (distilled water or 70% EtOH) in an Erlenmeyer flask and ultrasonicated for 15 min using an Elma Transsonics ultrasonic bath (Elma Schmidbauer GmbH, Singen, Germany) operated at 40 kHz ultrasonic frequency and 70 °C water bath temperature.

The extraction mixtures were further centrifuged and the supernatants were recovered; ethanolic extracts were supposed to the intermediary concentration under reduced pressure (using a rotary evaporator) in order to evaporate the ethanol. Finally, both aqueous and concentrated ethanolic extracts were freeze-dried using a BK-FD12S freeze dryer (Biobase Biodustry Co., Ltd., Shandong, China), eight dried extracts being obtained: QDRA M—*Q. dalechampii* rhytidome aqueous extract obtained through microwave-assisted extraction, QDRE M—*Q. dalechampii* rhytidome ethanolic extract obtained through microwave-assisted extraction, QDRA US—*Q. dalechampii* rhytidome aqueous extract obtained through ultrasound-assisted extraction, QDRE US—*Q. dalechampii* rhytidome aqueous extract obtained through ultrasound-assisted extraction, QFRA M—*Q. frainetto* rhytidome aqueous extract obtained through microwave-assisted extraction, QFRE M—*Q. frainetto* rhytidome ethanolic extract obtained through microwave-assisted extraction, QFRA US—*Q. frainetto* rhytidome aqueous extract obtained through ultrasound-assisted extraction and QFRE US—*Q. frainetto* rhytidome aqueous extract obtained through ultrasound-assisted extraction.

### 2.4. Quantification of Total Phenolic Compounds

Total phenolic content was measured based on a modified Folin-Ciocalteu method previously reported; the reaction mixture (100 µL extract, 100 µL Folin-Ciocâlteu reagent and 800 µL 5% sodium carbonate solution) was incubated for 30 min at room temperature, the absorbance being further read at 760 nm. Final results were calculated using a gallic acid calibration curve and expressed as mg gallic acid equivalents (GAE)/g bark. For the calibration curve 9 different gallic acid solutions were used. These solutions were all prepared from a 0.5% (*m/v*) gallic stock solution using water as a solvent. The concentrations were as follows: 0.05, 0.1, 0.15, 0.2, 0.25, 0.25, 0.3, 0.35, 0.4, 0.45 mg gallic acid/ mL of solution resulting in the linear equation: y = 11.767 * x + 0.2737. The freeze-dried extracts were redissolved in water reaching a final concentration of 0.4 mg freeze dried extract/ mL solution. Three samples were prepared for each extract. The absorbance of each sample was measured twice, resulting in 6 measurements. 

### 2.5. Quantification of Total Tannins

Total tannins content was measured according to the method described in European Pharmacopoeia 10.8 (tannins in herbal drugs) [12], as the difference between total phenolic content and phenolics non-absorbed by hide powder. The freeze-dried extracts (0.1 g) were redissolved in the same solvent (10 mL) used for extraction. The assay was performed in triplicate and the final results were expressed as percentage of pyrogallol.

### 2.6. UPLC-PDA Analysis of Individual Phenolic Constituents

For the identification of phenolic compounds, 0.02 g freeze dried extract was dissolved in 5 mL solvent (water or 70% EtOH), according to the solvent used for the extraction), and an aliquot of each sample was filtered through a 0.2 µm nylon membrane, before injection.

An UPLC Flexar FX-10 Perkin Elmer system, equipped with a binary pump, in-line degasser, autosampler, column thermostat, and a Flexar FX-PDA UHPLC detector was employed for the analysis of the individual phenolic compounds found in *Q. dalechampii and Q. frainetto* bark extracts. The method was previously described by Tanase et al. [25]; hence, chromatographic separation was achieved on a Luna C18 (2) column (3 µm particle size, 150 mm × 4.6 mm), with flow rate being set at 1 mL/min. Gradient elution with 0.1% formic acid (phase A)/acetonitrile (phase B) was operated as follows: 90% A and 10% B from 0.0 to 0.1 min, 90% to 20% A from 0.1 to 20.1 min, isocratic (20% A) from 20.1 to 25.1 min, followed by the increasing of A from 20% to 90% (25.1–26.1 min) and isocratic final segment (90% A from 26.1 to 30.1 min). Qualitative analysis was performed at four preferential wavelengths (270, 280, 324, and 370 nm respectively) using as reference substances chlorogenic acid, caffeic acid, gallic acid, epicatechin, catechin, eleuteroside B, quercetin, quercetin arabinoglycoside, vanillic acid, taxifolin, sinapic acid (20 µL of each standard solution (20 µg/mL) being supposed to injection).

### 2.7. Antioxidant Assays

Free radical-scavenging activity of the extracts was measured in a microplate reader (Epoch, BioTek, Winooski, VT, USA) using two in vitro assays, namely DPPH and ABTS, previously described in our work [24,26].

In the DPPH assay, serial dilutions of the extracts (100 µL) were mixed with a 0.1 mM DPPH solution (200 µL), the absorbance of the reaction mixture being read at 517 nm. Ascorbic acid was used as a positive control. Inhibitory capacity (IC%) was calculated using equation 1, with final results being expressed as IC_50_, (defined as the amount of the sample in µg/mL that scavenged 50% of the DPPH radical, determined by plotting the inhibition capacity against extract concentration):IC (%) = (A_DPPH_ − A_S_)/A_DPPH_ × 100,(1)
where A_DPPH_ is the absorbance of the control solution (DPPH solution with only solvent) and A_s_ is the absorbance of the sample.

For ABTS assay, serial dilutions of the extracts (100 µL) were mixed with a 10 mM ABTS solution (200 µL), the absorbance of the mixture being monitored at 734 nm. Trolox was used as a positive control. Inhibitory capacity (IC%) was calculated using equation 2, with final results being expressed as IC_50_, (defined as the amount of the sample in µg/mL that scavenged 50% of the ABTS radical, determined by plotting the inhibition capacity against extract concentration):IC (%) = (A_ABTS_ − A_S_)/ A_ABTS_ × 100,(2)
where A_ABTS_ is the absorbance of the control solution (ABTS solution with only solvent) and A_s_ is the absorbance of the sample.

For both DPPH and ABTS assays, the determinations were performed in triplicate.

### 2.8. In Vitro Enzyme-Inhibitory Potential

Inhibitory potential of the extracts against *α*-glucosidase, tyrosinase and acetylcholinesterase activity was measured using slightly modified microplate-adapted protocols previously described by Babotă et al. [27]; for each assay, the samples were previously dissolved in working buffer with 5% DMSO (10 mg/mL). Hence, for *α*-glucosidase inhibition, formation of *p*-nitrophenol was monitored at 405 nm after the reaction between serial dilutions of samples, yeast α-glucosidase (50 µL in phosphate buffer) and *p*-nitrophenyl-*α*-_D_-glucopyranoside (PNPG, 50 µL of 5 mM solution in phosphate buffer). Results were expressed in terms of IC_50_ using as positive control acarbose.

For tyrosinase assay, serial dilutions of each sample were previously prepared; 40 µL of each diluted sample were incubated in a 96-well plate with potassium phosphate buffer (80 µL, 50 mM, pH = 6.5) and mushroom tyrosinase solution (40 µL, 125 U/mL in phosphate buffer) for 5 min at 37 °C. Further, L-DOPA solution (40 µL, 10 mM in phosphate buffer) were added to the mixture and the plate was re-incubated for 15 min. Finally, the absorbance was measured at 492 nm, results being expressed in terms of IC_50_ using as positive control kojic acid.

The evaluation of acetylcholinesterase inhibition was done using a modified version of Ellman’s method; diluted sample (25 µL), Tris-HCl buffer (50 mM, pH = 8) was mixed with DTNB (125 µL, 0.9 mM Tris-HCl buffer) and acetylcholinesterase enzyme solution (25 µL, 0.078 U/mL) the mixture being further incubated at 37 °C for 15 min. Afterwards, ATCI (25 µL, 4.5 mM) was added and the plate was re-incubated for another 10 min at 37 °C. The absorbance of the final mixture was measured at 405 nm and results were expressed as IC_50_ value (µg/mL) using galantamine as a positive control.

### 2.9. Antimicrobial Activity Assessment

#### 2.9.1. Antimicrobial Activity Parameters (MIC, MBC and MFC)

The antimicrobial potential of the extracts was tested against five bacterial and three fungal strains (see Section 2) by evaluating minimum inhibitory concentrations (MIC), minimum bactericidal concentration, and minimum fungicidal concentration (MFC) [19,26].

For MIC measurement, a slightly modified microdilution method was employed [19,26]. First, bacterial suspensions were prepared by mixing 10 µL of 0.5 McFarland inoculum with 9990 µL of Muller–Hinton broth 2×. Further, each extract was dissolved at a concentration of 10 mg/mL (using purified water with 5% DMSO as solvent), the obtained solutions being filter-sterilized (0.2 µm PES syringe filters, Whatman Puradisc 25 mm). Finally, serial dilutions of each extract (200 µL) were pipetted in a 96-well plate together with 100 µL of bacterial suspension of each strain and incubated for 24 h at 37 °C. After incubation occurred, MIC values were optically evaluated.

MBC was assessed using 5 µL from 3 consecutive wells (from the MIC-well and from two more concentrated dilutions), which were sub-cultured on blood-agar plates and incubated for 24 h at 37 °C. MBC was defined as the dilution which did not show any bacterial growth on blood agar.

In a similar way, fungal inoculums were mixed with 0.5 McFarland and 9 mL of RPMI media buffered with MOPS and supplemented with 2% glucose. 100 µL of inoculum were treated with serial dilutions of each extract in order to measure MIC values (evaluated through microscopic examination).

In order to sustain the reproducibility of MIC method we performed additional quality control tests using gentamicin for bacterial strains and fluconazole for *Candida* spp., obtaining MIC values that are according with the standards.

#### 2.9.2. Inhibition of Biofilm Formation

As long as the selected bacterial strains are cited in literature as possessing the ability to produce biofilm, we evaluated the inhibitory potential of the extracts against biofilm formation using a previously developed method which uses RPMI medium (low-nutritional environment) [19].

Practically, in each well of a 95-well microplate were pipetted 100 µL of bacterial suspension (previously prepared by mixing 10 µL inoculum and 9990 µL RPMI medium) and 100 µL of diluted extract (calculated as 3%, 1.5%, 0.75%x concentration of previously tested MIC), the mixture being incubated at 37 °C for 18–24 h. Further, the plate was immersed in order to remove the excess RPMI and non-adherent cells, each well being treated after immersion procedure with crystal-violet aqueous solution (200 µL, 0.1% solution) and maintained in contact for 15 min. After an intermediary washing for the removal of the excess staining agent (three times immersion in purified water), the plate was dried and in each well were pipetted 200 µL of 30% acetic acid and incubated at room temperature (in order to solubilize the purple crystal adhered to the biofilm). Biofilm inhibition was expressed as an index according to the following Formula (3), using the absorbance measured for each well at 590 nm:BII = (TS − C)/CG,(3)
where BII means biofilm inhibition index, TS is the absorbance of the studied well, C is the absorbance of control, CG is the absorbance of control in growth. An index value below 0.75 was associated with the inhibition, while the values over 1.25 were associated with the stimulation. Values between 0.75 and 1.25 were attributed to the chance error.

Further, the index was converted into biofilm inhibition percentages by using Formula (4):% biofilm inhibition = (100 × BII) − 100,(4)Inhibition was considered for percentage values between −100% and −75%, while values greater than +25% were attributed to stimulation. Values between −25% and +25% were attributed to chance error.

#### 2.9.3. MRSA Adhesion on Sutures

Four sutures (A = Biosilk, 2/0, BX 544—natural silk, non-absorbable, multifilament, Biosintex, lot 13854; B = 1, Bicril, BX 146, polyglycolic acid, resorbable thread, multifilament, Biosintax, lot 13997; C = 1, Mono, BX W932, polyglecaprone 25, monofilament, absorbable, lot 19642; D = 1, Biopro, BX 3076, polypropylene, non-absorbable thread, monofilament, lot 14372) were incubated in the presence of the extracts, at a temperature of 28 °C for 24 h (to allow the extracts to adhere to the sutures). After the incubation, the sutures were dried for 1 h, at 50 °C. A 0.5 McFarland of MRSA inoculum was added over each dried suture, for 5 min. The sutures were afterward gently rinsed in sterile saline solution and then transferred in sterile tubes. Each sample was vortexed for 10 s. Fifty µL of each suspension were inoculated on blood agar and incubated at 36 °C for 24 h. The protocol was adapted by Dubas et al. [28]. The colonies were counted with the IUL Flash&Go colony counter. Sutures were also incubated without the extracts to serve as control samples. The experiment was performed in triplicate. The number of bacterial colonies obtained from the sutures impregnated with the *Quercus* extracts was compared with the number of bacterial colonies obtained by culturing the suspension from the control samples:% MRSA adhesion inhibition = ΔF/ΔR (5)
where ΔF = the number of MRSA colonies from the sutures treated with Quercus extracts; ΔR = the number of MRSA colonies from the control samples. An adhesion inhibition percentage value between −100% and −75% was considered inhibition, and a value higher than +25% was considered an enhancement of adherence (values between −25% and +25% were attributed to chance error). 

### 2.10. DNA Damage Assay

Comet assay method [29] was employed for the in vitro evaluation of DNA damage induced by the extracts against 293T human embryonic kidney cells; the assay was performed using CometAssay^®^ kit (3-well slides, ab 238544, Abcam Plc., Cambridge, United Kingdom). First, cells were cultivated in 24-well plates for 24 h in L15 medium, which after reaching 80% confluence were treated with each extract (3%, v/v, concentration of previously established MIC) for 1 h. In a second round, cells were cultivated from the beginning in the presence of extracts, for 24h in L15 medium. Two positive controls for damage DNA were created by treating the cells with hydrogen peroxide, respectively by exposure to UVC radiation for 1h. Negative control, without added substance, was also prepared. The cells were harvested by trypsinization, centrifuged, and then the sediment was washed with Mg^2+^ and Ca^2+^—free PBS and recentrifuged. The supernatant was discarded, the cells were resuspended in PBS (1 × 10^5^ cells/mL) and diluted 1:10 *v:v* with Comet Agarose preheated at 37 °C; 50 µL of diluted suspension were pipetted on CometSlide and allowed to solidify at 4 °C for 30 min. Cellular lysis was performed for 60 min at 4 °C in the dark using a pre-chilled Lysis Buffer provided in the kit, being followed by the treatment with Alkaline Solution in the same conditions. Finally, slides were supposed to electrophoresis procedure using a neutral buffer solution (provided with the kit) (1 h, at 4 °C, 1 v/cm voltage). After electrophoretic migration occurred, slides were rinsed subsequently with purified water and 70% ethanol; cells were further stained for 30 min by adding 100 µL of SYBR Green in each well. Finally, slides were rinsed again with PBS, dried at room temperature, and analyzed using an epifluorescence microscope.

### 2.11. Cytotoxic Effect

293T human embryonic kidney cells were used for the evaluation of the cytotoxic effect of the tested extracts. These cells are embryonic cells, highly active at 37 °C, the temperature of the human body. Cells cultivation was made in 24-well plates in L15 medium supplemented with FBS 10%, antibiotic-antimycotic 1% (10,000 U penicillin, 10 mg streptomycin, and 25 µg amphotericin B / mL) for 48 h. After reaching the semiconfluence, cells were washed, and a fresh medium containing the tested extracts (6%, 3%, 1.5%, and 0.25% in L15 media) was added and incubated for 24 h at 37 °C. After incubation, the cell growth was first documented by photography, then the supernatant from each well (containing also the detached, dead cells) was carefully discarded; the sediment consisting in still alive, adherent cells was treated with trypsin (80 µL per each sample) for 5 min, resuspended in 300 µL of L15 media supplemented with FBS 10% and centrifuged at 1500 rpm for 5 min. Finally, the supernatant was removed, the cells were re-suspended in PBS (50 µL) by vortexing and were enumerated under an optical microscope (10× magnification) in a counting chamber. The tests were performed in triplicate.

### 2.12. Statistical Analysis

All the assays were performed in triplicate. GraphPad Prism 8 was used for statistical analysis. The significance level was chosen before performing the statistical analysis (α = 0.05). Gaussian distribution of the analyzed data was confirmed through Kolmogorov–Smirnov normality test. Further, in order to compare the variance differences between the 2 groups, the F test was applied. The correct *t*-test was chosen accordingly, comparing the means of the two data series. One-way ANOVA, followed by Tukey post-hoc test was used for the analysis of TPC, TTC and antioxidant activity. A *p* value less than 0.05 was considered significant. For the assessment of the correlation between the TPC, antioxidant capacity and enzymatic inhibition the Pearson coefficients were calculated. 

## 3. Results

### 3.1. Phytochemical Profile of the Tested Extracts

The employed methods allowed us to estimate the occurrence of total phenolic and tannin fractions in *Q. dalechampii* and *Q. frainetto* barks extracts, the results being summarized in Figure 2. Important amounts of phenolic compounds were found in both species’ barks, the occurrence of these compounds being apparently influenced by the extraction method and solvents used for the preparation of the extracts. Hence, ethanolic extract obtained through microwave-assisted extraction from *Q. dalechampii* bark showed the highest phenolic content, followed by both aqueous and ethanolic extracts obtained through microwave-assisted extraction from *Q. frainetto*. A similar trend was observed for tannins content, the highest amounts being quantified in the extracts prepared using microwave-assisted method (QDRE M and QFRE M). Overall, the quantitative distribution of total phenolic compounds and tannins between *Q. dalechampii* and *Q. frainetto* barks seems to be similar.

The UPLC analysis of the phenolic compounds showed that all tested extracts contained gallic acid, catechin, and taxifolin (Table 1, Figure 3 and Figure 4). The vanillic acid was identified in the extracts obtained from *Q. dalechampii* bark (Figure 4). On the other hand, in extracts obtained from *Q. frainetto* bark, epicatechin, and caffeic acid were identified (Figure 3).

### 3.2. Antioxidant Potential

The results obtained after the evaluation of the free radical-scavenging activity of *Q. dalechampii* and *Q. frainetto* barks extracts are presented in Table 2. QFRE US was the most potent extract in the DPPH assay, while QDRE US was shown a higher radical-scavenging activity against ABTS•^+^. As it can be observed, the strongest neutralization of both radicals (DPPH• and ABTS•^+^) was achieved using the extracts prepared through ultrasound-assisted extraction, without important variations regarding the species collected. Conversely, the microwave-assisted treatment of the bark induced a slight to moderate decrease in the free radical-scavenging activity of the extracts, more visible in ABTS assay.

### 3.3. Inhibitory Potential against α-Glucosidase, Tyrosinase and Acetylcholinesterase

IC_50_ values measured for the extracts obtained from *Q. dalechampii* and *Q. frainetto* barks (summarized in Table 3) proved their ability to strongly inhibit α-glucosidase activity (better described by the inhibition curves presented in Figure 5) and an insignificant effect against tyrosinase and acetylcholinesterase enzymes.

Moreover, several differences regarding anti-glucosidase potential were observed between the tested extracts. The use of ethanol as extraction solvent slightly increased the inhibitory activity (proven by results obtained both for ultrasound- and microwave-assisted methods), while the influence of collecting species seemed to be also important (overall, the extracts of *Q. dalechampii* showed lower IC_50_ values in comparison with those ones obtained from *Q. frainetto* bark).

In Table 4, it can be observed that in the case of the microwave assisted extraction exists a moderate negative correlation between the total phenolic content and the ABTS radical scavenging activity. In contrast a very strong positive correlation was highlighted between the TPC and the neutralizing capacity of the DPPH radical, suggesting that a higher content of phenolics comprised in these extracts, indicate a higher antioxidant capacity only against the ABTS radical. Moreover, in the extracts obtained by microwave assisted extraction, was highlighted a moderate negative correlation between the phenolic content and the inhibition of the activity of α-glucosidase. This suggests that the microwave assisted extraction increases the likelihood of extracting α-glucosidase inhibitory compounds. 

In contrast, the ultrasound assisted extraction resulted in extracts with a weak to a moderate negative correlation between the phenolic content and the DPPH radical neutralization, while a weak positive correlation occurred between the TPC and the results of the ABTS assay. Additionally, a moderate negative correlation was highlighted between the phenolic content and the inhibition of acetylcholinesterase. This indicates a higher potential of the ultrasounds to extract more acetylcholinesterase inhibiting compounds.

### 3.4. Antimicrobial and Antibiofilm Activity

MIC and MBC values obtained after the assessment of the antibacterial potential of *Q. dalechampii* and *Q. frainetto* barks extracts (Table 5) suggested a mild to moderate effect against the Gram-positive strains (*S. aureus, MRSA*) and a low effectiveness against Gram-negative ones (*E. coli*, *K. pneumoniae*, *P. aeruginosa*); at the same time, *E. coli* was found as being resistant to all tested extracts (CMI and CMB values greater than 5 mg/mL indicate the lack of sensitivity). Conversely, *S. aureus* was found as the most sensible strain, especially to the ethanolic extracts obtained through microwave-assisted extraction (QDRE M and QFRE M), while MIC values were comparable for *MRSA* and *K. pneumoniae* after exposure to all extracts.

Antifungal activity was tested against three *Candida* species, all the extracts being found as inactive against *C. albicans* and *C. parapsilopsis* (Table 6). A weak inhibition was obtained for *C. krusei* after exposure to the extracts obtained through microwave-assisted extraction.

Considering the results obtained in the evaluation of MIC, MBC, and MFC, antibiofilm activity assessment was further performed for *Q. dalechampii* and *Q. frainetto* bark extracts. The tested extracts inhibit the biofilm formation ability of MRSA, while QDREM, QDRAU, QFREM, QFRAU (but not higher concentrations of QDREU and QFREU) enhance the biofilm formation for *E. coli*. QDRAU and QFRAU stimulated the biofilm formation for *K. pneumoniae*. Additionally, the extracts have inhibitory effects on *P. aeruginosa* biofilms, but only in 3% concentrations (Table 7). 

The *Quercus* extracts enhanced the adherence of MRSA on the studied suture material, at different degrees (Table 8), depending on the suture.

### 3.5. Cytotoxicity and Damage DNA Testing for Human Embryonic Kidney Cells

The effects of *Quercus* extracts on human cells showed that only QFREM induced damage of cellular DNA, as seen in Figure 6. No damaged cells were found in presence of the other extracts.

The direct cytotoxic effect was assessed after incubation of either attached cells in culture or fresh cells, in presence of different concentrations of QDREM or QFREM. The results showed that both extracts were able to exert important cytotoxic effects at 6% and 3% concentrations, on confluent cells. In concentration of less than 1.5%, the cytotoxic effect was limited, but the number of living cells decreased if compared with the control. If the fresh cells were cultivated in presence of *Quercus* extracts, they were not allowed to attach and grow, regardless of the extract type and concentration (Table 9). Microscopical findings of treated cell culture with QDREM and QFREM are exemplified in Figure 7. It can be seen that the morphology of cells was altered especially in 6% concentration.

## 4. Discussion

The obtained results reveal the *Q. dalechampii* and *Q. frainetto* barks as valuable by-products, which can be used as an alternative source for the recovery of several phenolic constituents (mainly phenolic acids and tannins). As could be observed in Figure 1, no major differences could be distinguished between these two *Quercus* species barks regarding the quantitative distribution of total phenolic and tannins contents. Otherwise, different treatments applied to the plant material induced several important changes regarding the amounts of bioactive compounds found in the extracts. The obtained data describe an increased recovery rate both for total phenolic compounds and tannins after the microwave-assisted treatment. Moreover, the use of hydroethanolic solvent allowed a better extractability for the aforementioned compounds, including the tannins. Even though tannins are described as water-soluble phytoconstituents, our results suggest an improved recovery for these compounds by using microwave-assisted extraction with 70% EtOH as solvent. These results are in line with the last findings regarding the extraction of the tannins from plant matrices, summarized in a review study by Atanu Kumar et al. [30].

A minimum of 3% tannin content is the first quality criterion defined by the pharmacopeial regulations for the herbal product *Quercus cortex* [12]. In this regard, both *Q. dalechampii* and *Q. frainetto* barks could be considered as potential substitutes for the pharmacopeial product, considering that our results showed values between 27–40% total tannins in the dry raw material extracted through different methods. At the same time, the barks of the aforementioned species seem to contain higher amounts (almost two times higher) of phenolic compounds than the *Q. robur* bark, for which Bouras et al. [31] reported TPC values between 1.56 and 2.14 g/100 g dry product, also extracted through microwave-assisted extraction (in comparison with our findings—2.89 to 3.72 g/100 g dry product).

Regarding the individual phenolic constituents’ distribution in the analyzed samples, the results obtained after UPLC-PDA assessment are in line with those previously reported by Galiñanes et al. [32]. Using an RP-HPLC-ESI-TOF method, the authors confirmed the presence of gallic acid, ellagic acid, and (+)-catechin in *Q. frainetto* bark aqueous extracts, as well as for two glycosylated flavonoid derivatives (kaempferol 3-glucoside and quercetin 3-O-rhamnoside). To the best of our knowledge, no previous reports on the phenolic profile of *Q. dalechampii* bark were published; hence, the presence of gallic acid, catechin, taxifolin, and vanillic acid as main constituents of *Q. dalechampii* bark is mentioned for the first time in the present study.

The phytochemical profile of the analyzed samples influenced their bioactivity, as well as the extraction method. The important amounts of tannins found in QDRE M and QFRE M extracts seem to be responsible for their cytotoxic potential against 293T human embryonic kidney cells, as long as only these extracts exerted this bioactivity. Conversely, the antioxidant assays revealed the extracts obtained through ultrasound-assisted extraction as the most potent free radical-scavenging agents, a similar inconsistent trend being described for the *α*-glucosidase inhibition assay. These results could be explained probably due to the different distribution of individual phenolic constituents in the analyzed samples, which could not be reported at the moment (data under processing). It was previously reported that the inhibitory potential of *Quercus mongolica* Fisch. ex Ledeb cups against *α*-glucosidase activity were dependent on the ellagic acid content quantified in the extracts obtained from this species [33]; a similar activity was observed for persimmon (*Diospyros kaki* Thunb.) fruits extract, which exerted anti-*α*-glucosidase activity depending on the amounts of a condensed-tannin type derivative specific for this fruit [34].

Tannins are recognized as valuable antimicrobial agents, intensively promoted as alternative or complementary options to classic antibiotics-based therapies [35,36,37]. As could be observed in Table 5 and Table 6, the lowest MIC, MBC, and MFC values were obtained for the tannin-rich extracts, QDRE M and QFRE M, respectively. In terms of selectivity of the antibacterial potential, our findings are described by the same trend previously reported in our assessment regarding *Q. rubra* [19], for which was also observed a higher affinity for Gram-positive than for Gram-negative ones. As well as for *Q. rubra*, *Q. dalechampii* and *Q. frainetto* barks seem to be more active as antibacterial agents instead of antifungals. In the carried experiments, the *Quercus* extracts did not inhibit the adhesion of MRSA and *K. pneumoniae*. We cannot be sure if the results are due to the fact that the extracts have no effect on bacterial adhesion, or if the impregnation of the suture was not optimal for the sutures’ materials. The literature describes various protocols for the impregnation of suture materials [38,39], and more studies should be performed to find out the most efficient impregnation method.

As well as for antimicrobial potential, tannins are cited as promising biomolecules with cytotoxic and antitumor properties [40,41,42]. Several assays were developed in order to evaluate the capacity of different substances to induce cell death, one of the most known being the “comet” assay [43,44]. This test is based on the ability of negatively charged loops/fragments of DNA to be drawn through an agarose gel in response to an electric field, electrophoretic migration inducing a ‘comet’-like shape to damaged DNA, size, and shape of the comet, as well as the distribution of DNA within the comet being correlated with the intensity of DNA damage [44]. In the present study, comet assay was used as a complementary test in the assessment of genotoxic effects of *Q. dalechampii* and *Q. frainetto* barks extracts, revealing a weak ability of this preparation to interact with 293T human embryonic kidney cells DNA, observed just for the cells exposed to QFRE M. This result was completed by those obtained after the evaluation of direct cytotoxic effect against the same cell line, which could not be observed.

## 5. Conclusions

*Q. dalechampii* and *Q. frainetto* bark extract obtained through microwave- and ultrasound-assisted extraction were evaluated in terms of chemical and bioactive profile. The obtained results confirmed the presence of tannins as main phenolic constituents, especially for the ultrasound-assisted preparations, which were found as the most active in terms of antioxidant and anti-*α*-glucosidase activity. A mild antibacterial and cytotoxic potential were proven for the same preparations, the obtained results opening further perspectives regarding the quantitative evaluation of the individual phenolic constituents of these by-products. This study demonstrated that the *Q. dalechampii* and *Q. frainetto* bark extracts would be a promising material in the development of a new drug for its outstanding antidiabetic, antioxidant, and antimicrobial activities. It is important, after identifying the biological potential of these extracts, to determine the specific molecule responsible for the given activity. Thus, the above characterizations can be further completed by identifying specific metabolites or active molecules present in the evaluated extracts. It is also necessary to describe the interactions between the bioactive compounds and the biological pathways of action.

## Figures and Tables

**Figure 1 pharmaceutics-15-00343-f001:**
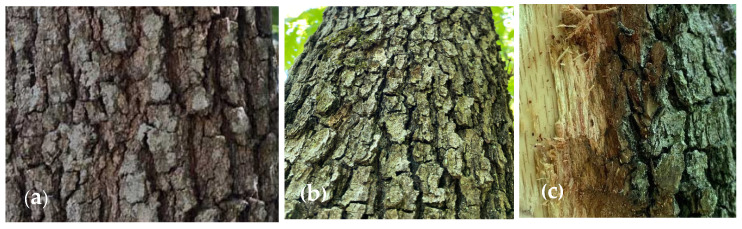
The morphological aspect of (**a**) *Q. dalechampii* and (**b**,**c**) *Q. frainetto* bark.

**Figure 2 pharmaceutics-15-00343-f002:**
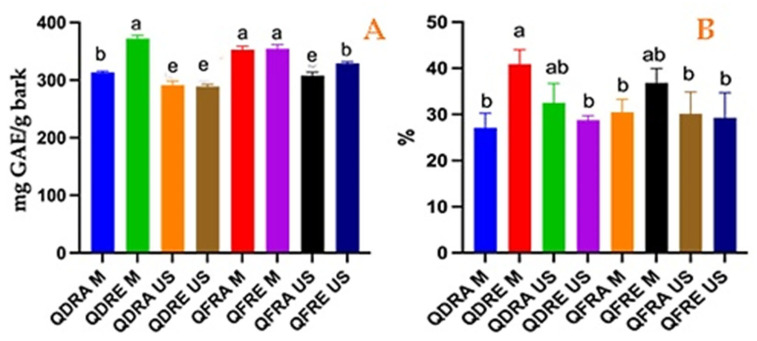
Total phenolic content (**A**) (mg GAE/g bark) and total tannin content (**B**) (expressed as %) quantified in *Q. dalechampii* and *Q. frainetto* barks extracts. Different lower-case letters indicate significant differences between extracts.

**Figure 3 pharmaceutics-15-00343-f003:**
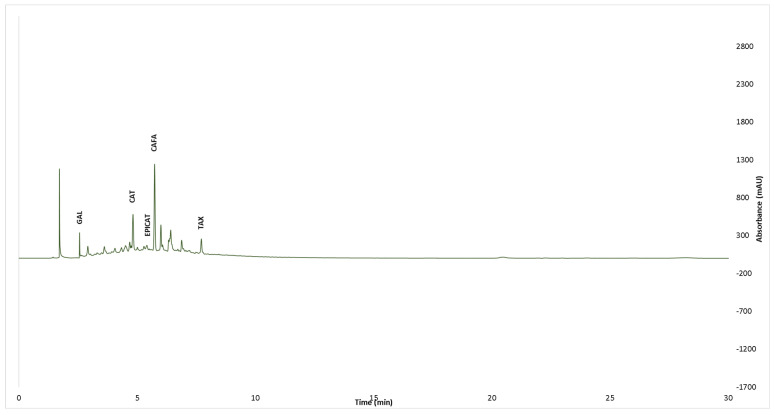
HPLC chromatogram of *Q. frainetto* bark extract at 280 nm.

**Figure 4 pharmaceutics-15-00343-f004:**
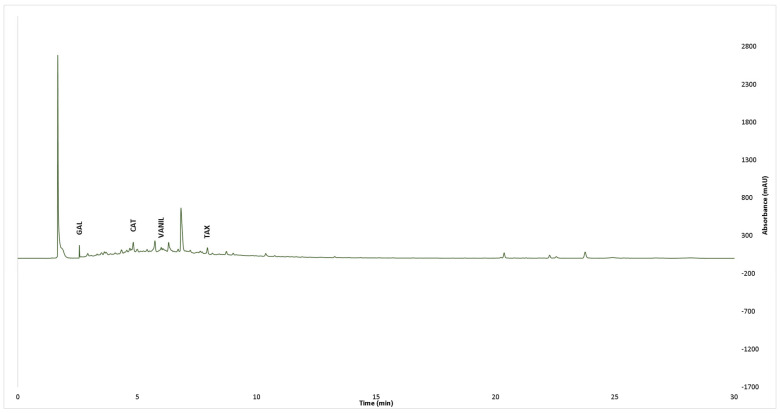
HPLC chromatogram of *Q. dalechampii* at 280 nm.

**Figure 5 pharmaceutics-15-00343-f005:**
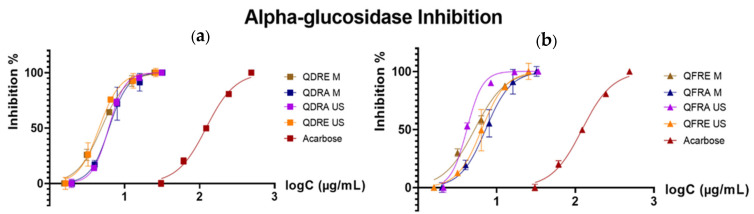
Graphs indicating the dependence between the logC (expressed as the concentration in terms of µg/mL) and the inhibition percentage for the tested samples—*Q. dalechampii* (**a**) and *Q. frainetto—*(**b**) bark extracts, in the case of α-glucosidase inhibitory activity, along with acarbose as positive control.

**Figure 6 pharmaceutics-15-00343-f006:**
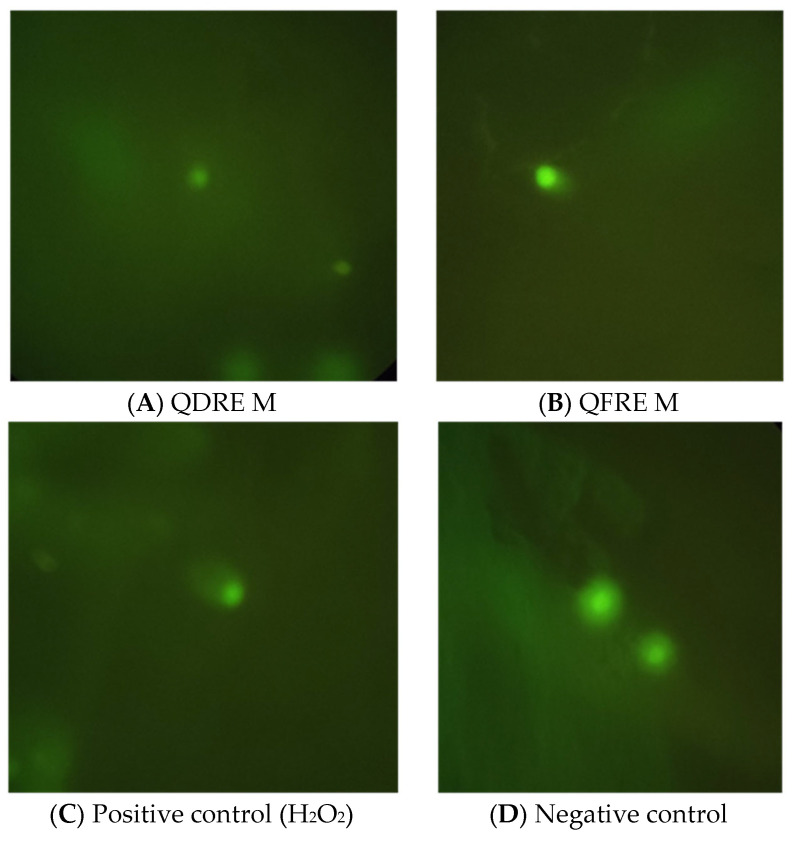
Representative images showing the DNA damage effects. (**A**) Cells treated with QDREM, negative result. (**B**) Cells treated with QFREM, positive result, showing the “comet tail” following the DNA damage. (**C**) Positive result in presence of H_2_O_2_. (**D**) Negative control showing non-damaged cells; visualization using 40× objective.

**Figure 7 pharmaceutics-15-00343-f007:**
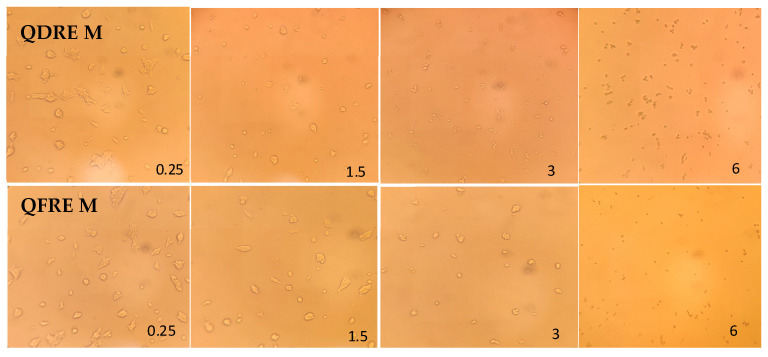
Microscopic features of 293T human embryonic kidney cells after the evaluation of cytotoxic effects of QDRE M and QFRE M extracts tested at 0.25, 1.5, 3, and 6% of MIC (visualization using 20× objective).

**Table 1 pharmaceutics-15-00343-t001:** Retention time of the identified compounds.

Compound	Abbreviation	Retention Time
Gallic acid	GAL	2.8 min
Catechin	CAT	4.8 min
Taxifolin	TAX	7.7 min
Caffeic acid	CAFA	5.7 min
Epicatechin	EPICAT	5.5. min

**Table 2 pharmaceutics-15-00343-t002:** Antioxidant potential of *Q. dalechampii* and *Q. frainetto* barks extracts.

Sample	IC_50_ DPPH (µg/mL)	IC_50_ ABTS (µg/mL)
QDRA M	2.27 ± 0.109 ^e^	8.453 ± 0.138 ^a^
QDRE M	9.153 ± 0.418 ^a^	7.856 ± 0.779 ^a^
QDRA US	3.86± 0.33 ^d^	2.49± 0.045 ^c^
QDRE US	2.8± 0.379 ^e^	2.194± 0.1 ^c^
QFRA M	8.036 ± 0.435 ^b^	6.556 ± 0.553 ^b^
QFRE M	9.399 ± 0.517 ^a^	6.135 ± 0.261 ^b^
QFRA US	4.07± 0.54 ^cd^	2.48±0.046 ^c^
QFRE US	2.66± 0.31 ^e^	2.37± 0.1 ^c^
Ascorbic acid	0.0044 ± 0.0002 ^f^	-
Trolox	-	1.06 ± 0.44 ^d^

Different letters in the same column mean statistically significant differences at *p* < 0.05.

**Table 3 pharmaceutics-15-00343-t003:** Enzyme-inhibitory potential of *Q. dalechampii* and *Q. frainetto* barks extracts.

Enzyme	Species	Extract/ Positive Control	IC_50_ (µg/mL)
α-Glucosidase	*Q. dalechampii*	QDRE M	4.91
		QDRA M	6.21
		QDRE US	4.58
		QDRA US	6.12
	*Q. frainetto*	QFRE M	5.24
		QFRA M	7.24
		QFRE US	6.44
		QFRA US	4.17
		Acarbose	122.27
Acetylcholinesterase	*Q. dalechampii*	QDREM	133.4
		QDRAM	139.2
		QDREU	137.4
		QDRAU	220.7
	*Q. frainetto*	QFREM	147.3
		QFRAM	188.1
		QFREU	136.1
		QFRAU	170.7
		Galantamine	0.0002
Tyrosinase	*Q. dalechampii*	QDREM	106.00
		QDRAM	125.16
		QDREU	67.2
		QDRAU	224.00
	*Q. frainetto*	QFREM	131.32
		QFRAM	147.02
		QFREU	82.04
		QFRAU	353.8
		Kojic acid	4.44

**Table 4 pharmaceutics-15-00343-t004:** Pearson’s correlation coefficient (r) among TPC, DPPH and ABTS (antioxidant capacity) and α-Glucosidase, acetylcholinesterase and tyrosinase (enzymatic inhibition) for tested extracts.

	TPC	DPPH	ABTS	α-Glucosidase	Acetylcholinesterase	Tyrosinase
Extracts obtained by microwave assisted extraction						
**TPC**	1.000	0.946	−0.462	−0.418	0.071	−0.256
**DPPH**	0.946	1.000	−0.713	−0.360	0.188	−0.034
**ABTS**	−0.462	−0.713	1.000	−0.123	−0.577	−0.609
**α-Glucosidase**	−0.418	−0.360	−0.123	1.000	0.846	0.827
**Acetylcholinesterase**	0.071	0.188	−0.577	0.846	1.000	0.887
**Tyrosinase**	−0.256	−0.034	−0.609	0.827	0.887	1.000
Extracts obtained by ultrasounds assisted extraction						
**TPC**	1.000	−0.320	0.198	0.395	−0.454	−0.089
**DPPH**	−0.320	1.000	0.798	−0.325	0.789	0.952
**ABTS**	0.198	0.798	1.000	0.215	0.739	0.794
**α-Glucosidase**	0.395	−0.325	0.215	1.000	0.195	−0.422
**Acetylcholinesterase**	−0.454	0.789	0.739	0.195	1.000	0.585
**Tyrosinase**	−0.089	0.952	0.794	−0.422	0.585	1.000

**Table 5 pharmaceutics-15-00343-t005:** MIC and MBC values (expressed as mg/mL), which describe the antibacterial activity of *Q. dalechampii* and *Q. frainetto* barks extracts.

Species	Sapmple/ Control	*S.aureus*		*MRSA*		*E. coli*		*K. pneumoniae*		*P. aeruginosa*	
		MIC	MBC	MIC	MBC	MIC	MBC	MIC	MBC	MIC	MBC
*Q. dalechampii*	QDRA M	0.31	2.5	0.62	0.62	>5	>5	0.62	0.62	5	5
	QDRE M	0.31	0.31	0.62	0.62	>5	>5	0.62	1.25	0.62	2.5
	QDRA US	2.5	0.62	0.62	0.62	>5	>5	0.62	0.62	2.5	5
	QDRE US	0.31	1.25	0.62	0.62	>5	>5	0.62	0.62	1.25	2.5
*Q. frainetto*	QFRA M	0.31	5	0.62	1.25	5	5	0.62	0.62	2.5	5
	QFRE M	0.31	0.16	0.62	0.62	>5	>5	0.62	2.5	0.62	2.5
	QFRA US	0.62	0.62	0.62	0.62	>5	>5	0.62	0.62	1.25	>5
	QFRE US	0.31	0.62	0.62	0.62	>5	>5	0.62	2.5	1.25	5

**Table 6 pharmaceutics-15-00343-t006:** MIC values (expressed as mg/mL) which describe the antibacterial activity of *Q. dalechampii* and *Q. frainetto* barks extracts.

Species	Sapmple/ Control	*C. albicans*	*C. parapsilosis*	*C. krusei*
		MIC	MIC	MIC
*Q. dalechampii*	QDRA M	>5	>5	5
	QDRE M	>5	>5	2.5
	QDRA US	>5	5	>5
	QDRE US	>5	5	2.5
*Q. frainetto*	QFRA M	>5	>5	2.5
	QFRE M	>5	>5	5
	QFRA US	>5	>5	5
	QFRE US	>5	>5	5

**Table 7 pharmaceutics-15-00343-t007:** Biofilm inhibition (expressed in %) of *Q. dalechampii* and *Q. frainetto* barks extracts on *S. aureus*, *MRSA*, *E. coli*, *K. pneumoniae* and *P. aeruginosa*; the values marked in bold represent significant changes compared to control.

			Biofilm Inhibition				
Sample		Concentration	*S. aureus*	MRSA	*E. coli*	*K. pneumoniae*	*P. aeruginosa*
	QDREM	3	6.2	**−43.1**	**49.0**	18.6	**−48.1**
*Q. dalechampii*		1.5	0	**−44.0**	**48.3**	17.4	6.4
		0.75	2.1	**−45.7**	**65.8**	16.8	−6.6
	QDRAU	3	19.0	**−35.1**	**102.7**	**44.3**	−23.9
		1.5	11.8	**−40.1**	**68.2**	**32.3**	−5.3
		0.75	7.7	**−39.2**	**52.4**	**26.9**	**35.9**
	QDREU	3	−3.6	**−40.1**	16.8	20.4	**−45.7**
		1.5	−6.2	**−41.3**	20.5	20.4	−4.5
		0.75	−8.7	**−42.2**	**75.7**	16.8	−5.6
	QFREM	3	−2.1	**−41.3**	**55.8**	15.6	**−49.7**
*Q. frainetto*		1.5	−2.6	**−43.1**	**43.2**	10.2	−1.6
		0.25	0.5	**−51.9**	**57.5**	11.4	−5.1
	QFRAU	3	17.4	**−34.8**	**64.7**	**41.9**	**−35.1**
		1.5	12.3	**−38.1**	**59.6**	**35.3**	14.9
		0.75	8.7	**−40.7**	**64.7**	**29.3**	13.3
	QFREU	3	−7.2	**−45.4**	8.2	8.4	**−50.3**
		1.5	−14.4	**−43.7**	11.3	7.2	−6.1
		0.75	−13.3	**−43.4**	**37.0**	10.2	−11.2

**Table 8 pharmaceutics-15-00343-t008:** *MRSA* adherence (expressed in %) on sutures treated with QDREM and QFREM, compared with control samples; the values marked in bold represent significant changes compared to control.

Suture	QDREM	QFREM
A	**+66**	+16
B	**+95**	**+31**
C	**+99**	+17
D	**+30**	−0.5

**Table 9 pharmaceutics-15-00343-t009:** The cytotoxic effect of *Q. dalechampii* and *Q. frainetto* barks extracts.

Sample/Control	Concentration	Average Number of Cells after Treatment of Attached, Confluent Cells in Presence of *Quercus* Extracts for 24 h	Average Number of Cells after Cultivation of Non-Attached, Fresh Cells in Presence of *Quercus* Extracts for 24 h
QDRE M	6%	0 ± 0	0 ± 0
	3%	1 ± 1	0 ± 0
	1.5%	30.7 ± 4.5	0 ± 0
	0.25%	25.3 ± 4.5	0 ± 0
QFRA M	6%	0 ± 0	0 ± 0
	3%	1.3 ± 1.2	0.3 ± 0.6
	1.5%	50.3 ± 1.5	0 ± 0
	0.25%	46 ± 6.2	0 ± 0
H_2_O_2_	-	150 ± 15.6	9.7 ± 0.6

## Data Availability

Not applicable.

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
