# Peer review of "Potential Use of *Quercus dalechampii* Ten. and *Q. frainetto* Ten. Barks Extracts as Antimicrobial, Enzyme Inhibitory, Antioxidant and Cytotoxic Agents"

_pharmaceutics, 2023, doi:10.3390/pharmaceutics15020343_

Round 1

Reviewer 1 Report

Further Perspectives Regarding the Use of Quercus dalechampii and Q. frainetto Barks for the Development of Extracts with Antimicrobial, Enzyme Inhibitory, Antioxidant and Cytotoxic Potential

The authors carried out extensive work with extracts of Quercus, however, they failed in some experiments and should be reviewed. In this form it cannot be accepted in the journal.

The abstract is very long, please shortened it to a maximum of 200 words. The language must also be reviewed. For example, the first paragraph (lines 32-35) is confusing.

The keywords are not suitable and should be changed and focus on more relevant aspects of the study.

In section 2.2., the authors must attach the morphological characteristics including images previously evaluated to authenticate the species under study.

In section 2.3, specify how many extractions were performed on the same sample in each method? If it was done only once, how do you ensure that the extraction was efficient?

In section 2.4, specify the concentrations of gallic acid used to make the calibration curve. Specify the extract concentrations used for the samples. It should also be specified in which solvents the samples were dissolved for UV-vis analysis. If necessary, the calibration curves must be attached. Specify how many times the experiment was repeated.

In section 2.5. Same comment as in section 2.4. Specify concentrations and solvents for each case. How many times was the experiment repeated?

In section 2.6. Specify the concentrations used for the samples and the solvent used.

In section 2.7. Specify the calibration curve used for both DPPH and ABTS. Why is it not compared to a positive control, eg vitamin C, Trolox, etc.? Specify how many times the experiment was repeated.

The header of section 2.8.1 is so long. Shorten it.

In section 2.8 the authors evaluated the antimicrobial activity, but the authors failed in the experiment by not using a positive control to make a comparison with any commercial antibiotic or fungicide, this is necessary for any pharmacological study. Specify how many times the experiment was repeated.

Line 332, specify from where or how the 293T cell line was obtained.

In figure 1, you must add subsections to differentiate one figure from another. It's confusing. Also, the foot of the figure must be rewritten in another way, it is very repetitive.

In figure 1 (phenolics) there is an error in the homogeneous groups between the letters “a” and letters “d”.

In Table 1 there are inconsistencies with the homogeneous groups for DPPH. The authors must be careful in the analyses.

In all the tables the footer is so repetitive, please rewrite them or restructure all the tables.

The foot of the figures is so long and needs to be modified.

Author Response

Dear reviewer,

First of all, thank you for the professional comments and observations. We thank you for your comments, which have made us think carefully about our data sets again. Accordingly, we have reanalyzed these where necessary. Please find below our point-by-point itemized answer and correction. We write to say that we now strongly believe that we can convince you that the data is sound and that we have adequately answered various valid concerns.

Response to reviewer 1

  1. The abstract is very long, please shortened it to a maximum of 200 words.

Author’s Response: Thank you! Revised as requested.

  1. The language must also be reviewed. For example, the first paragraph (lines 32-35) is confusing.

Author’s Response: Thank you! Revised as requested!

  1. The keywords are not suitable and should be changed and focus on more relevant aspects of the study.

Author’s Response: Thank you! The keywords list was updated.

  1. In section 2.2., the authors must attach the morphological characteristics including images previously evaluated to authenticate the species under study.

Author’s Response: Thank you! We have included the required information and representative figures.

  1. In section 2.3, specify how many extractions were performed on the same sample in each method. If it was done only once, how do you ensure that the extraction was efficient?

Author’s Response: Thank you! As we specified in the Statistical analysis section, all the determinations (including extraction) were performed in triplicate.

  1. In section 2.4, specify the concentrations of gallic acid used to make the calibration curve. Specify the extract concentrations used for the samples. It should also be specified in which solvents the samples were dissolved for UV-vis analysis. If necessary, the calibration curves must be attached. Specify how many times the experiment was repeated.

Author’s Response: Thank you! We have included the required information in section 2.4.

  1. In section 2.5. Same comment as in section 2. Specify concentrations and solvents for each case. How many times was the experiment repeated?

Author’s Response: Thank you! The mentioned details were added.

  1. In section 2.6. Specify the concentrations used for the samples and the solvent used.

Author’s Response: Thank you! The concentrations and the type of solvent used were added.

  1. In section 2.7. Specify the calibration curve used for both DPPH and ABTS. Why is it not compared to the positive control, eg vitamin C, Trolox, etc.? Specify how many times the experiment was repeated.

Author’s Response: Thank you! We initially performed the determinations for positive controls (ascorbic acid for DPPH assay and Trolox for ABTS assay), but we did not consider it relevant to introduce these results in the manuscript. However, we have introduced the IC50 for positive controls in table 1, and the method description was updated.

  1. The header of section 2.8.1 is so long. Shorten it.

Author’s Response: Thank you! Revised as requested!

  1. In section 2.8 the authors evaluated the antimicrobial activity, but the authors failed in the experiment by not using a positive control to make a comparison with any commercial antibiotic or fungicide, this is necessary for any pharmacological study. Specify how many times the experiment was repeated.

Author’s Response: Thank you for this comment! In our opinion, it has no sense to compare the effect of a commercial antibiotic/antifungal on standard ATCC strains, as these are well characterized, and their susceptibility is well known. Moreover, for example, S. aureus ATCC 43300 is MRSA (thus resistant to all beta-lactams), while S. aureus ATCC 25923 is MSSA. Similarly, C. albicans is sensitive to fluconazole, C. krusei is resistant. The study showed the antimicrobial effect of the extracts at a concentration of mg/mL. The antimicrobial experiments were performed in one round. The MIC method is highly reproducible and standardized for example by EUCAST or CLSI (even the in vitro diagnostic methods do not require triplicate for this).

  1. Line 332, specify from where or how the 293T cell line was obtained.

Author’s Response: Thank you! The cells were due to the courtesy of the Institute of Virology, University of Zurich, Switzerland.             

  1. In figure 1, you must add subsections to differentiate one figure from another. It's confusing. Also, the foot of the figure must be rewritten in another way, it is very repetitive.

Author’s Response: Thank you! We introduced some changes in this section.

  1. In figure 1 (phenolics) there is an error in the homogeneous groups between the letters “a” and letters “d”.

Author’s Response: Thank you for this valuable observation! Revised as requested!

  1. In Table 1 there are inconsistencies with the homogeneous groups for DPPH. The authors must be careful in the analyses.

Author’s Response: Thank you for this valuable observation! Revised as requested!

  1. In all the tables the footer is so repetitive, please rewrite them or restructure all the tables.

Author’s Response: Thank you! We modified the footer.

  1. The foot of the figures is so long and needs to be modified.

Author’s Response: Thank you! We modified the footer.

Reviewer 2 Report

The manuscript by Tanase et al. focuses on Quercus by-products as a potential source of bioactive compounds. I appreciate their efforts in studying these materials, which are waste products. However, the article lacks many details, and the actual final application of these kinds of extracts missed me. Many issues need to be solved:

Abstract: LC-MS/MS should be removed.

Par. 2.6: How were the samples prepared for UPLC analysis?

Par. 2.8 In vitro enzyme inhibitory potential: How were the extracts dissolved for analysis?

There are two "2.8" paragraphs.

Par. 2.10: Why did the authors test the cytotoxic effect on HEK 293T cells? Is there a specific reason for using this cell model? What is the aim of the cytotoxic effect on these cells? The concentration of the extract is unclear: is the percentage v/v, w/v?

Par. 3.1: The authors titled the paragraph "phytochemical profile," but no profile is visible in that paragraph. They only showed the results of total polyphenol and tannin content. They should report chromatograms at least one for Q. dalechampii and one for Q. frainetto bark extracts and the rest as supplementary material. A table with the name of the identified compounds and retention time is also needed. Since the authors have standards of phytochemicals, they can perform a quantitative analysis of these compounds in the extracts to improve the comparison.

Par. 3.2: How do you explain that for all samples the IC50 related to DPPH is higher than that of ABTS, except for sample QDRA M where the IC50 of ABTS is higher than that of DPPH?

Par. 3.3 Check the caption of Table 2.

Par. 3.5 Check the caption of Figure 3.

The authors could perform a correlation analysis between the polyphenol/tannin content (or specific compounds) and the proposed activities to see if the activity correlates with a specific class of compounds.

Author Response

Dear reviewer,

First of all, thank you for the professional comments and observations. We thank you for your comments, which have made us think carefully about our data sets again. Accordingly, we have reanalyzed these where necessary. Please find below our point-by-point itemized answer and correction. We write to say that we now strongly believe that we can convince you that the data is sound and that we have adequately answered various valid concerns.

Response to reviewer 2

  1. Abstract: LC-MS/MS should be removed.

 Author’s response: Thank you! We introduced some changes in this section.

  1. Par 2.6 How were the samples prepared for UPLC analysis?

Author’s Response: Thank you! The materials and methods section was updated with the required information.

  1. 2.8 In vitro enzyme inhibitory potential: How were the extracts dissolved for analysis?

Author’s response: Thank you! We updated this section with the requested information.

  1. There are two "2.8" paragraphs.

Author’s response: Thank you! The numbering was updated.

  1. 2.10: Why did the authors test the cytotoxic effect on HEK 293T cells? Is there a specific reason for using this cell model? What is the aim of the cytotoxic effect on these cells? The concentration of the extract is unclear: is the percentage v/v, w/v?

Author’s response: Thank you! 293T cells are embryonic cells, highly active at 37C, the temperature of the human body. Also, the kidney is one of the human body filters, highly exposed to toxic effects of substances. The concentration is v/v.

  1. 3.1: The authors titled the paragraph "phytochemical profile," but no profile is visible in that paragraph. They only showed the results of total polyphenol and tannin content. They should report chromatograms at least one for Q. dalechampii and one for Q. frainetto bark extracts and the rest as supplementary material. A table with the name of the identified compounds and retention time is also needed. Since the authors have standards of phytochemicals, they can perform a quantitative analysis of these compounds in the extracts to improve the comparison.

Author’s response: Thank you! Representative chromatograms were added for QD and QF extracts. Also, the table with the identified compounds and Rt was added.

  1. 3.2: How do you explain that for all samples the IC50 related to DPPH is higher than that of ABTS, except for sample QDRA M where the IC50 of ABTS is higher than that of DPPH?

Author’s response: Thank you! Although it was an unexpected result, It is not an unusual phenomenon. We have a large experience with these type of determinations, and we've found that there aren't necessarily strong relationships between the two approaches. The most plausible explanation is related to the phytochemical profile of the extracts and the solvents used for extraction, as both can greatly influence the antioxidant activity. The ABTS radical preferably reacts via the SPLET (sequential proton loss electron transfer) mechanism in aqueous solutions, whereas the DPPH radical preferably reacts via the SPLET mechanism in solvents such as ethanol and methanol. Also, the reactions are greatly influenced by the pH. In conclusion, although the results do not have a good correlation, it could be interesting in the future to search and find the factors that influence to such a great extent, the response of ABTS and DPPH radicals.

  1. 3.3 Check the caption of Table 2.

Author’s response: Thank you! We corrected the caption of Table 2.

  1. 3.5 Check the caption of Figure 3

Author’s response: Thank you! We corrected the caption of Table 2.

  1. The authors could perform a correlation analysis between the polyphenol/tannin content (or specific compounds) and the proposed activities to see if the activity correlates with a specific class of compounds.

Author’s response: Thank you! We have included the required information.

Round 2

Reviewer 1 Report

The authors made most of the corrections, however, the justification for the failure on the controls in their antimicrobial experiments is still not clear. Authors are required to strongly justify this part or perform additional experiments.

In my previous comment my question was because the authors did not use a positive control? And they mention that it doesn't make sense because the susceptibility is well known.

Now my questions are: How can the authors evaluate the antibacterial potential of their extracts if they do not have a control to compare?

Good effectiveness should hover in terms of micrograms per milliliter (less than 50 µg/mL). The authors obtained results at concentrations as high as 300 mµ/mL.

In addition, it must be considered that this Journal is specialized in Pharmaceuticals and the controls must not go unnoticed. See some examples:

https://doi.org/10.3390/pharmaceutics14091952

https://doi.org/10.3390/pharmaceutics14040698

Author Response

Dear reviewer,

We thank you for your comments, which have made us think carefully about our data sets again. Accordingly, we have reanalyzed these where necessary. Please find below our answer and correction. We write to say that we now strongly believe that we can convince you that the data is sound and that we have adequately answered various valid concerns.

Response to reviewer 1

  1. The authors made most of the corrections, however, the justification for the failure on the controls in their antimicrobial experiments is still not clear. Authors are required to strongly justify this part or perform additional experiments. In my previous comment, my question was because the authors did not use a positive control. And they mention that it doesn't make sense because the susceptibility is well-known. Now my questions are: How can the authors evaluate the antibacterial potential of their extracts if they do not have a control to compare? Good effectiveness should hover in terms of micrograms per milliliter (less than 50 µg/mL). The authors obtained results at concentrations as high as 300 mµ/mL.

Author’s Response: Thank you so much for your suggestions! We assure you of our appreciation, and we performed more experiments hoping you will find our work worthy of publication.

To assess the antimicrobial activity, we determined the MIC (minimum inhibitory concentration) of the tested substances, according to the EUCAST guidelines (https://www.eucast.org/fileadmin/src/media/PDFs/EUCAST_files/Disk_test_documents/2022_manuals/Reading_guide_BMD_v_4.0_2022.pdf)

For antibiotics, the MIC is interpreted according to the breakpoints available (there are specific breakpoints for certain antibiotics, breakpoints that are different for each bacterial/fungal species). In the present study, we tested the effects of our substances, hence, there are no breakpoints available. This is another element of novelty in the present study. The experiments prove that there is an antimicrobial effect, which is later on quantified based on the MIC value. The aim of the study is to assess if the tested substances have or have not antimicrobial effects, not to compare the potency of those antimicrobial effects with the effect of antibiotics.

Keeping in mind the reviewers' suggestions, we performed additional control tests using gentamicin and fluconazole, obtaining MIC values of:

  • Gentamicin ( aureus – 0.25 mg/mL, MRSA – 32 mg/mL, E. coli – 0.25 mg/mL, K. pneumoniae – 0.5 mg/mL, P. aeruginosa – 0.5 mg/mL)
  • Fluconazole ( albicans – 0.125 mg/mL, C. parapsilosis – 1 mg/mL, C. krusei – 8 mg/mL)

Also, we have growth control serving as a positive control and ensuring that the experiment is valid from a microbiological point of view.  The initial experiments, combined with the new ones, strengthen the validity of our results, and we hope you will consider our work satisfactory.

Reviewer 2 Report

I believe that the changes made by the authors resulted in a high improvement of the manuscript. The paper is now suitable for publication in its present form.

Author Response

We would like to thank you for your valuable comments and suggestions that helps us to improve the quality of the present mnuscript.

Round 3

Reviewer 1 Report

The authors considered most of the observations and it can be accepted only with some additional modifications:

There is no clear objective in the Abstract or in the Introduction section. Please clarify it in both sections.

Figures 3, 4 and 5 improve resolution.

Table 6, 7 and 8. Eliminate the "%" in all values. It is enough that this is placed in the table header.

Author Response

Thank you for your valuable recommendations!

We have implemented the requested modifications.